# Microporosities in 3D-Printed Tricalcium-Phosphate-Based Bone Substitutes Enhance Osteoconduction and Affect Osteoclastic Resorption

**DOI:** 10.3390/ijms21239270

**Published:** 2020-12-04

**Authors:** Chafik Ghayor, Tse-Hsiang Chen, Indranil Bhattacharya, Mutlu Özcan, Franz E. Weber

**Affiliations:** 1Center of Dental Medicine, OC, Oral Biotechnology & Bioengineering, University of Zurich, 8032 Zurich, Switzerland; chafik.ghayor@usz.ch (C.G.); chen.robert.nh@gmail.com (T.-H.C.); Indranil.Bhattacharya@usz.ch (I.B.); 2Center for Surgical Research, University Hospital and University of Zurich, 8091 Zurich, Switzerland; 3Center of Dental Medicine, Division of Dental Biomaterials, Clinic for Reconstructive Dentistry, University of Zurich, 8032 Zurich, Switzerland; mutlu.ozcan@zzm.uzh.ch; 4CABMM, Center for Applied Biotechnology and Molecular Medicine, University of Zurich, 8057 Zurich, Switzerland

**Keywords:** microporosity, osteoconduction, microarchitecture, nanoarchitecture, bone substitute, additive manufacturing, 3D printing, ceramics

## Abstract

Additive manufacturing is a key technology required to realize the production of a personalized bone substitute that exactly meets a patient’s need and fills a patient-specific bone defect. Additive manufacturing can optimize the inner architecture of the scaffold for osteoconduction, allowing fast and reliable defect bridging by promoting rapid growth of new bone tissue into the scaffold. The role of scaffold microporosity/nanoarchitecture in osteoconduction remains elusive. To elucidate this relationship, we produced lithography-based osteoconductive scaffolds from tricalcium phosphate (TCP) with identical macro- and microarchitecture, but varied their nanoarchitecture/microporosity by ranging maximum sintering temperatures from 1000 °C to 1200 °C. After characterization of the different scaffolds’ microporosity, compression strength, and nanoarchitecture, we performed in vivo studies that showed that ingrowth of bone as an indicator of osteoconduction significantly decreased with decreasing microporosity. Moreover, at the 1200 °C peak sinter temperature and lowest microporosity, osteoclastic degradation of the material was inhibited. Thus, even for wide-open porous TCP-based scaffolds, a high degree of microporosity appears to be essential for optimal osteoconduction and creeping substitution, which can prevent non-unions, the major complication during bone regeneration procedures.

## 1. Introduction

Osteoconduction is defined as a process of ingrowth of sprouting capillaries, perivascular tissue, and osteoprogenitor cells from a bony bed into the 3D structure of a porous implant (from Cornell and Lane [1,2]) used as a cue to guide bridging of a defect in bony tissue [3]. Early observations of osteoconduction were based on transplantation of autologous bone. The initial approaches to define the ideal osteoconductive microarchitecture were undertaken in the 1990s, utilizing scaffolds with randomly distributed pores and channel-based microarchitectures to mimic autologous bone. Based on those findings, pore diameters of 0.3–0.5 mm have long been falsely regarded as optimal for osteoconduction [4,5].

The production of a library of bone substitute scaffolds with defined pore sizes, pore locations, and defined connections between pores by additive manufacturing has reportedly identified scaffolds with pores between 0.7 mm and 1.2 mm to be highly osteoconductive [6]. Scaffolds with pore diameters of up to 0.5 mm and beyond 1.5 mm were found to be far less osteoconductive, in that they displayed substantially decelerated defect bridging, which is an indirect measure of osteoconduction [6]. A cranial defect model [7,8] has been used to determine osteoconduction in numerous studies [6,9,10,11,12]. The low mechanical challenge posed in the cranium is an advantage for testing diverse micro- and nanoarchitectures, since it allows for testing without the need for costly fracture fixation devices, which is indispensable in long bone defect treatments. Moreover, cranial defects are clinically highly relevant in congenital anomalies, trauma, stroke, aneurysms, and cancer [13].

Additive manufacturing, in contrast to subtractive manufacturing, is a methodology that builds objects layer by layer. In the context of regenerative medicine, additive manufacturing by diverse methods (reviewed in [14]) will develop into a key technology in the realization of personalized medicine. The production of bone substitutes by 3D printing has been extensively reviewed [15,16,17]. The architecture of such personalized bone substitutes has been defined on three levels. Macroarchitecture represents the millimeter- to centimeter-sized outer shape of the scaffold. If macroarchitecture is identical to the dimensions of the bone defect, the bone substitute is patient specific. Microarchitecture, on the other hand, describes the micrometer- to millimeter-sized inner spatial distribution of the material, defining pore size, shape, porosity, channels, and pore interconnectivity. The surface and inherent porosity of the material at the nano-to-micrometer scale is defined as nanoarchitecture [3]. Nanoarchitecture is predominantly dependent on post-processing methodology, including sintering regimes and surface treatments. To a lesser extent, it also depends on the underlying additive manufacturing procedure used.

Microporosity and surface roughness are specified at the next architectural level, that of nanoarchitecture. Several groups have reported bone ingrowth and the presence of cells in micropores well below 0.1 mm in diameter [18,19,20]. During bone regeneration, microporosity mainly affects the extent of osseointegration of the implant. The extent of bone regeneration, however, appears to be independent of microporosity, according to tests with TCP-based scaffolds containing macropores 0.15 mm in diameter [21]. The sizes of the macropores in the scaffolds used in this study are far too small to be osteoconductive. Therefore, the contribution of nanoarchitecture, in the form of microporosity, to the osteoconductivity of scaffolds with wide-open, porous, highly osteoconductive microarchitectures remains elusive. To that end, we produced a series of highly osteoconductive TCP-based scaffolds with identical, wide-open microarchitectures through additive manufacturing, in a range of microporosities by varying maximal sintering temperature during post-processing. The objective of this study was to evaluate the contribution of microporosity to the osteoconductivity of wide-open porous scaffolds and to determine if the microporosity of TCP-based wide-open porous scaffolds is a key factor in osteoconduction. Incidentally, we found that a lack of microporosity inhibits degradation of TCP-based scaffolds by osteoclasts.

## 2. Results

### 2.1. Scaffold Characterization

After sintering, scaffolds appeared white, irrespective of the applied sintering temperature, suggesting that all yellowish binder material was removed during the sintering process (Figure 1 upper panel). Scanning electron microscopy (Figure 1b) revealed that upon sintering, TCP grains grow in a temperature-dependent manner from 0.71 ± 0.15 µm at 1000 °C to 3.08 ± 1.56 µm at 1200 °C and undergo partial fusion.

Pore diameter is also affected by increasing sintering temperature. Grain and pore sizes of the partially sintered scaffolds are listed in Table 1.

### 2.2. Microporosity of TCP Based Scaffolds

Overall, microporosity was calculated as proportional to shrinkage in all three axes, and was determined experimentally by weight gain upon infiltration with distilled water. Theoretical and experimental values (Table 1) were almost identical, indicating that by loading the test sample from one side, as depicted in Figure 2a, H_2_O could fill up the entire microporous capacity of the test samples to a depth of 4.0 mm within 15 min.

### 2.3. Compression Strength of Partially Sintered Scaffolds

The compression strengths of sintered scaffolds with the same microarchitectures we later used to perform in vivo osteoconduction tests were determined. Only the diameter of the connections was increased from 0.7 mm to 1.0 mm to be in line with our standard microarchitecture for mechanical testing. With increase in sintering temperature, compression strength increased from 1.50 ± 0.37 N/mm^2^ (mean ± S.E.M.) at 1000 °C, to 2.25 ± 0.84 N/mm^2^ at 1100 °C, and 7.81 ± 2.12 N/mm^2^ at 1200 °C. Only the scaffolds sintered at 1100 °C and 1200 °C fell within the naturally occurring range of cancellous bone (Figure 2b) [22] of 2–12 N/mm^2^.

### 2.4. Osteoconductivity of Microporous Scaffolds In Vivo

One aim of this study was to determine the relationship between osteoconductivity and degree of microporosity for a highly osteoconductive scaffold with open porous microarchitecture. The microarchitecture of all scaffolds was identical because they were built with the same stl-file, and the printer software incorporated compensatory dimensions in x, y, and z for sintering temperature-dependent shrinkage. The histology of sections from the middle of each scaffold (Figure 3a) revealed that over a 4 week period, bone formation and advancement of bone into the scaffold varied with sintering temperature. The extent of bony bridging and bone formation in the defect was quantitated (Figure 3b,c). In defects treated with scaffolds sintered at 1000 °C, 85.26 ± 19.16 % of the middle section was bridged, 64.68 ± 23.40 % was bridged in those sintered at 1100 °C, and 50.15 ± 18.33% was bridged in those sintered at 1200 °C. In the area of interest, the percentage of bony regeneration in the middle section was 74.45 ± 18.84% with scaffolds sintered at 1000 °C, 52.60 ± 19.95 % for those sintered at 1100ºC, and 33.66 ± 12.53% for those sintered at 1200 °C. For both measures, scaffolds sintered at 1000 °C performed significantly better than scaffolds sintered at either 1100 °C or 1200 °C. These results suggest that the higher microporosity found in scaffolds sintered at 1000 °C compared to scaffolds sintered at 1100 °C or 1200 °C enhances bony bridging and bone regeneration.

### 2.5. Ion Release from Partially Sintered Scaffolds

The maximal sintering temperature had a profound effect on scaffold surfaces (Table 1) and possibly changed the release of ions. To assess this possibility, we determined Ca^2+^ and (PO4)^3−^ ion release from the three scaffold types, which revealed a decrease in calcium ion release from scaffolds sintered at higher temperatures (Figure 4a). Ca^2+^ release over 60 days was 8.77 ± 0.32 µmol/g from scaffolds sintered at 1000 °C, 7.79 ± 0.22 µmol/g from scaffolds sintered at 1100 °C, and 5.65 ± 0.16 µmol/g from scaffolds sintered at 1200 °C. Differences in (PO4)^3−^ ions released over a 60-day period were marginal (Figure 4b): 9.88 ± 0.90 µmol (PO4)^3−^ ions released/g from scaffolds sintered at 1000 °C, 9.29 ± 1.36 µmol (PO4)^3−^ ions released/g from scaffolds sintered at 1100 °C, and 9.89 ± 1.06 µmol (PO4)^3−^ ions released/g from scaffolds sintered at 1200 °C.

### 2.6. Osteoclastic Resorption of Microporous Scaffolds In Vitro

The final stage of implant remodeling is creeping substitution, in which the implant is gradually replaced by newly formed bone. The overall degradability of the scaffold is the sum of its dissolution due to ion release (Figure 4) and cell-based resorption. We next assessed the influence of sintering temperature and microporosity on osteoclastic resorption modelled with RAW264 cells. We failed to find osteoclasts on the surface of highly microporous scaffolds sintered at 1000 °C (Figure 5a,d,g), since seeded cells quickly disappeared into the scaffold, where they defied detection by electron microscopy. On scaffolds sintered at 1100 °C and 1200 °C, RANKL-induced differentiation of RAW cells gave rise to osteoclasts on scaffold surfaces (Figure 5e,f). Incubation of scaffolds seeded with RAW264 cells altered pit resolution on the surface of scaffolds sintered at 1000 °C and 1100 °C (Figure 5g,h). No changes were apparent on the surfaces of scaffolds sintered at 1200 °C (Figure 5i). The most impressive effect on surface morphology and number of resorption pits was seen on scaffolds sintered at 1100 °C and exposed to osteoclastic cells (Figure 5l) compared to identically sintered but unexposed surfaces (Figure 5k).

## 3. Discussion

In this report, we generated tricalcium-phosphate-based scaffolds using 3D printing, and assessed the influence of microporosity on bone formation and osteoconduction in vivo, and on osteoclastic resorption in vitro. We kept the material and microarchitecture constant and varied different aspects of the nanoarchitecture by increasing the maximum sintering temperature (Table 1). These studies revealed that osteoconduction and bone regeneration in cranial defects treated with scaffolds containing identical wide-open porous microarchitecture were significantly improved when, during post-processing, scaffolds were sintered at a low peak sintering temperature of 1000 °C (Figure 3), giving them a high microporosity of 39%.

Tuning the sintering temperature affects the nanoarchitecture of a scaffold in many ways. Therefore, parameters including grain size, micropore diameter, and overall microporosity that cannot be studied independently [21,23] varied across our three scaffold types as well (Table 1). Recently, decreased grain size was shown to enhance bone formation [24,25]. Others have shown that scaffolds with a fixed macropore diameter of 0.15 mm, grain size of 1.3 µm, and microporosity of 10% performed the same in terms of defect healing as scaffolds with a grain size of 3.3 µm and microporosity of 25% [21]. This is in keeping with our results on osteoconduction and bone regeneration, since we also detected no significant difference in these parameters when comparing scaffolds sintered at 1100 °C with a microporosity of 22% and a grain size of 1.2 µm with those sintered at 1200 °C with a microporosity of 1% and a grain size of 3.1 µm. Only scaffolds sintered at 1000 °C with 39% microporosity displayed a significant increase in osteoconduction and bone regeneration. The need for microporosities of at least 32% to enhance bone formation in rats has been shown by others [26], whose data support our finding that 39% but not 22% microporosity enhances osteoconduction and bone formation. The drawback to a low sintering temperature is lower mechanical strength (Figure 2b). This correlation has been previously demonstrated [27]. Increasing mechanical strength to the level of cancellous bone without changing the current microarchitecture (Figure 2b) can easily be achieved by increasing the wall thickness of the unit cell. After compressive loading, failure type analysis indicated that the origin of failures was either related to connector morphology upon initial exposure to compressive loading, or to small defects at the most external corner of the specimen, possibly contributing to propagation of cracks towards the connectors. Hence, improvement in connector dimensions and morphology and flawless fabrication (especially at the edges of the structure) may enhance overall strength, and needs to be further investigated.

In the aforementioned work [21], the macropore diameter was 0.15 mm, which was very different from the highly osteoconductive pore diameter range of 1.0–1.2 mm [6]. This applies to all previous studies of microporosity in which assembled granules or 3D-printed scaffolds with rod distances less than 0.4 mm were used [28]. One major functional difference between these two macropore size ranges is that bone formation in scaffolds with pores smaller than 0.4 mm occurs predominantly on the scaffold surface. At macropore diameters of 0.5 mm and beyond, bone formation can occur within the channel, distant from the scaffold surface [29]. In wide-open porous microarchitectures, as we used here, initial bone formation occurs distant from the surface of the scaffold, as has been independently shown for titanium and TCP-based scaffolds [3,9,11]. This unique spatial situation, in which bone formation occurs in wide-open porous scaffolds, minimizes effects on osteoconduction imposed by additional nanoarchitectural parameters [24] including surface roughness, grain size, surface wettability, protein absorption, and direct interaction of bone forming cells with the surface. Therefore, our observations may provide insights into how microporosity as an isolated factor affects osteoconduction irrespective of the additional nanoarchitectural parameters mentioned above. Moreover, by applying our recently identified highly osteoconductive microarchitecture (with pore diameters of 1.2 mm and connection diameters of 0.7 mm) [6] characterized by extremely fast bone ingrowth distant from the scaffold surface, this study focuses on the impact of microporosity on optimal osteoconductivity.

A recent review summarized possible mechanisms by which microporosity may influence bone formation [30]. First, micropores can provide niches for cells that preferentially undergo osteogenic differentiation and cells can be attracted to settle in micropores by capillary forces [20]. Second, micropores enhance waste removal and nutrient supply. The ready flow of water, which carries water-soluble nutrients and wastes, and its occupation of the entire interconnected micropore system is illustrated in Figure 2a, and is supported by the fact that calculated microporosity based on shrinkage is identical to microporosity determined by water uptake (Table 1). Third, microporosity increases the surface area of the scaffold and allows tuning of ion release during bone regeneration. For wide-open porous scaffolds, such as we used here, the niche aspect does not apply, since bone formation occurs at a significant distance from the surface [9,11]. Significant improvement in waste removal and nutrient supply due to increased microporosity in this microarchitecture is unlikely, since the microarchitecture is already wide-open and porous, allowing unrestricted fluid flow throughout the scaffold. Moreover, the area where bone formation occurs is spatially separated from the microporous material. This notion was further supported by a comparison of identical wide-open porous lattice architectures produced from titanium devoid of micropores, or with TCP, which contains micropores [9]. Both promoted osteoconduction and bone formation equally well. Therefore, micropores as such are not essential in wide-open porous structures to yield high osteoconductivity [9,11]. Assessment of the third potential mechanism (i.e., that microporosity increases surface area and ion release) yielded the conclusion that reducing sintering temperature and increasing microporosity increases specific surface area 5- to 36-fold (Table 1) and increases Ca^2+^ release in a time-dependent fashion by 1.7- to 2.6-fold (Figure 4a). If we reference Ca^2+^ ion release by scaffold volume instead of weight, the amount of released calcium is higher over the first 10 days, which is the most relevant time for bone regeneration [31], by a factor of 1.2- to 1.5-fold. This suggests that bone ingrowth as a surrogate measure for osteoconduction and bone regeneration is accelerated by highly microporous scaffolds at the molecular level by an increase in Ca^2+^ concentration sufficient to act as an osteoinductive signal.

Osteoinduction has for many years been directly associated with bone morphogenetic proteins and bone formation at ectopic sites [2,32,33]. Early reports on osteoinduction with ceramic implants were published in the 1990s [34,35]. In 2002, it was discovered that biphasic calcium phosphate can induce bone formation at ectopic sites and displays osteoinductive characteristics in vivo [36] beyond a geometric osteoinductive potential [37]. That calcium-phosphate-based osteoinduction can be tuned by the sinter temperature and is associated with the nanoarchitecture has also been reported [38], and a direct link to calcium ion release and its effect on diverse cell types involved in bone regeneration has been summarized [39]. The minimal specific surface area required for osteoinduction by calcium phosphates was found to be 1 m^2^/g [39]. Only our scaffolds sintered at 1000 °C meet this threshold (Table 1). This further supports our current conclusion that highly osteoconductive microarchitectures facilitate accelerated bone ingrowth via osteoinduction mediated by elevated Ca^2+^ ion release from the TCP scaffold (Figure 4a). Mesenchymal stem cells seeded on these types of scaffold upregulate BMP-2 3-fold and produce an osteoinductive stimulus [40]. Since bone formation occurs at a depth within the scaffold beyond this nanoarchitectural feature, the effect of increased microporosity is indirect, and is presumably mediated by elevated Ca^2+^ ion levels acting directly on cells involved in bone formation without any direct contact with this nanoarchitectural feature. For mechanically weak calcium phosphates, such accelerated bone ingrowth is essential, since the newly formed bone in the scaffold will increase the mechanical strength of the scaffold/bone composite early in the healing process. Shortening the time required to establish a solid, mechanically stable bone bridging reduces the risk of the main complication of bone regeneration procedures, i.e., manifestation of a non-union [41].

The type of osteoinduction reported here, enhancement of osteoconduction by scaffolds with wide-open porous microarchitectures, is distinct from osteoinduction by material-induced heterotopic ossification [42]. In contrast to osteoinduction enhanced by elevated Ca^2+^ release, material-induced heterotopic ossification is only seen months after implantation. Bone formation starts in the midst of the scaffold, occurs on surfaces and in micropores of the scaffold, and occurs heterotopically. Material-induced bone formation is associated with reduced Ca^2+^ levels as a result of calcium deposition into the apatite layer forming on the surface of the scaffold and due to lack of vascularization [42]. Since none of these characteristics apply to our wide-open porous scaffolds, and high macroporosity would be detrimental to material-induced heterotopic ossification, the enhanced osteoconduction mediated by increased Ca^2+^ release falls into another category of osteoinduction.

Following consolidation of the scaffold and treated defect by bony bridging, creeping substitution must occur by gradual degradation of the scaffold. We thus studied the effect of varying microporosity on degradation of our scaffolds, both by chemical dissolution (physicochemical degradation) (Figure 4) and resorption (cellular degradation by osteoclasts) (Figure 5). The most striking difference in resorption was seen between scaffolds sintered at 1100 °C and 1200 °C. Since the surfaces of scaffolds without microporosity were devoid of resorption pits, they were apparently not resorbed by osteoclastic cells at all (Figure 5). In this setting, nanoarchitectural features like microporosity cannot be distinguished independently from surface roughness, grain size, surface wettability, and protein absorption, because osteoclasts interact directly with the surface. Beyond nanoarchitectural characteristics, high calcium ion concentrations inhibit osteoclastic resorption and mediate osteoclast detachment [43,44,45]. In our array of scaffolds, however, those with the lowest calcium release were not resorbed by osteoclasts, which rule out a negative effect of calcium in our system. Another possible nanoarchitectural feature on the surface of scaffolds that could influence resorption is grain size (Table 1). An increase in grain size decreases osteoclastic resorption of TCP [46]. This dependency on grain size could explain why the surfaces of scaffolds sintered at 1200 °C with a grain size of 3.08 µm are less susceptible to osteoclastic resorption than those sintered at 1100 °C with a grain size of 1.24 µm. Increasing grain size inversely correlates with length of grain boundaries. Since resorption is most intense at grain boundaries [47], this correlation suggests that a decrease in grain boundary length could reduce or halt resorption.

Osteoclastic resorption plus dissolution determines the overall rate of scaffold degradation, which is fastest in scaffolds sintered at lower temperatures. The dependence of resorption on nanoarchitectural characteristics could explain some conflicting results in the field, where some groups report that synthetic TCP does not undergo resorption by RAW cells differentiated to osteoclasts [48], while others observe resorption [47]. We found both to be true, and showed that the resorbability of TCP by osteoclasts depends on the sintering temperature imposed. The decisive factor for TCP resorbability appears to be sintering temperature-dependent changes in nanoarchitecture.

## 4. Materials and Methods

### 4.1. Implant Production

The overall microarchitecture of the scaffolds was chosen from a library in which we had earlier identified highly osteoconductive pore-based microarchitectures [6]. The scaffold was composed of unit cells formed by cubes 1.5 mm in length. A 1.2 mm pore is located in the center of each unit cell. The pore in each unit cell is connected to all six sides of its cube by centrally located cylinders with a diameter of 0.7 mm. The assembly of unit cells to form the final implant is illustrated in Figure 6 and was reported earlier [6,11].

The TCP scaffolds were produced with TCP slurry LithaBone™ TCP 300 (Lithoz, Vienna, Austria) as previously reported [9]. In brief: The CeraFab 7500 system (Lithoz, Vienna, Austria) was used to solidify the slurry by exposure of each 25 µm layer of the photoactive polymer to a blue LED light at a resolution of 50 µm in the x/y-plane. The green body of the scaffold was then produced in a layer-by-layer fashion [49]. After the green body was removed from the building platform of the printer with a razor blade, it was cleaned with LithaSol 20™ (Lithoz, Vienna, Austria) and pressurized air. The polymeric binder needed to keep the ceramic particles in place was decomposed during a thermal treatment regime, followed by sintering to increase the density of the ceramic particles. The sintering procedure incorporated a dwell time of 3 h at 1000 °C, 1100 °C, or 1200 °C to tune overall microporosity by incomplete sintering. Differences in green body shrinkage due to different maximal sintering temperatures were compensated for by adjusting the dimensions of the green body in all three axes so that the macro- and microarchitecture after sintering were identical. The sintered scaffolds were transferred onto a sterile bench, packed for incorporation into the operation workflow, and used as bone substitute implants without further sterilization.

### 4.2. Scanning Electron Microscopy (SEM)

Scaffold examinations were performed at a service lab at the University of Zurich using a Zeiss Supra V50 scanning electron microscope (SEM) (Carl Zeiss, Oberkochen, Germany). Scanning was performed under an acceleration voltage of 12 kV, with a distance between sample and detector of 9.5 cm. Based on these images, grain size and pore size were determined.

### 4.3. Microporosity

The degree of microporosity was calculated based on the empirically determined shrinkage of the green body in all three dimensions during post-processing. Samples sintered at 1200 °C were considered to have a microporosity of 0%, as evident in Figure 1. In addition, we determined microporosity based on uptake of distilled water into a test scaffold composed of an upper circular platform (5.0 mm in diameter, 2.5 mm thickness) and a lower circular platform (4.0 mm in diameter and 1.5 mm thickness). Examinations were performed at a service lab at the University of Zurich using a Zeiss Supra V50 scanning electron microscope (SEM) (Carl Zeiss, Oberkochen, Germany).

### 4.4. Surgical Procedure

Eight full grown (6–8 months old) female New Zealand white rabbits were used to examine osteoconduction by the different scaffolds using a previously reported calvarial defect model [12]. Animal weights ranged from 3.5 to 4.0 kg, and all animals were fed a standard laboratory diet. The procedures were evaluated and accepted by the local authorities (114/2015 and 065/2018) and are in line with the EU Directive 2010/63/EU for animal experiments. Prior to surgery, animals were anesthetized by injection of 65 mg/kg ketamine and 4 mg/kg xylazine. Anesthesia was maintained with isoflurane/O_2_. After disinfecting the site, an incision was made from the nasal bone to the mid-sagittal crest, soft tissues were deflected, and the periosteum was removed. Next, four evenly distributed 6-mm-diameter cranial bone defects were created with a trephine bur under copious irrigation with sterile saline in the operation field. All defects were completed with rose burrs of 5 mm diameter, followed by a burr with a 1 mm diameter to preserve the dura. The defects were flushed with saline solution, and the implants were applied by gentle press fitting. Each animal received all three treatment modalities. Treatment modalities were assigned to random positions in the first animal, and thereafter, cyclically permuted clockwise. Treatments were grouped according to sintering temperature (1000 °C, 1100 °C, and 1200 °C) and were labeled 1000, 1100, or 1200, respectively. Four weeks after the operation, rabbits received general anesthesia and were sacrificed by an overdose of pentobarbital. The cranial section containing all four craniotomy sites was removed and placed in 40% ethanol. Methacrylate-embedding was performed as previously reported [11].

### 4.5. Histomorphometry

A ground section from the middle of each implant was evaluated using image analysis software (Image-Pro Plus®; Media Cybernetic, Silver Springs, MD, USA). The area of interest (AOI) was defined as the 6 mm defect width and the area fraction of the implant submerged into the bony defect. The area of new bone in the AOI was determined to be the percentage of bone and bony integrated scaffold in the AOI (bony area, %).

### 4.6. Bone Bridging

Quantitation of bony bridging as a measure of osteoconduction was performed as previously reported [50,51]. Areas with bone tissue within the defect margin were projected onto the *x*-axis. Next, all stretches of the *x*-axis with projecting bone tissue were summed up and divided by the defect width of 6 mm. Bone bridging is given as a percentage of the defect width (6 mm) where bone formation had occurred.

### 4.7. Compression Strength Measurements

Specimens (7.5 mm × 7.5 mm × 6.0 mm) were mounted in the jig of the Universal Testing Machine (Zwick ROELL Z2.5 MA 18-1-3/7, Ulm, Germany) where the occlusal surface (6.0 mm × 7.5 mm perpendicular to the orientation of the individual layers) was subjected to compressive loading with a crosshead speed of 1 mm/min. The stress–strain curve was analyzed with TestXpert software (TestXpert V11.02, Zwick ROELL, Ulm, Germany). To characterize the failure mode, specimens were examined using an optical microscope (Zeiss MC 80 DX, Jena, Germany) at 40X magnification, followed by examination with a digital microscope at 300X to 1000X magnification (Keyence, Osaka, Japan).

### 4.8. Osteoclast Differentiation on a TCP Scaffold and Resorption Pit Assay

RAW264.7 cells were cultured in DMEM supplemented with 10% FBS and antibiotics (100 units/mL penicillin G and 100 mg/mL streptomycin). The cultures were never allowed to become confluent. Incubations were performed at 37 °C in 5% CO_2_ in humidified air. Then, 2.0 × 10^4^ RAW264.7 cells were seeded onto each scaffold. The scaffolds were placed at 37 °C for 2h to allow cells to adhere followed by the addition of a complete growth medium. The growth medium was replaced with a fresh medium every 2 days to eliminate non-adherent cells. Cells were scraped from culture flasks, seeded onto 3D-printed scaffolds, and allowed 4h to become adherent. Scaffolds were then transferred to new 6-well plates (one scaffold/well) and further cultured in the presence or absence of 50 ng/mL RANKL. Medium was replaced every 2 days for a period of 9 days of stimulation. Cell morphology and resorption lacunae were evaluated by scanning electron microscope (SEM). Briefly, after incubation, samples were rinsed with 0.1 M phosphate-buffered saline and fixed with 2.5% glutaraldehyde solution (SIGMA) overnight at 4 °C. Fixed samples were then dehydrated in an ethanol series (30%, 50%, 70%, 80%, 90%, 95%, and 100%), followed by a critical point drying procedure. Samples were then gold coated and observed under SEM for osteoclast morphology. For resorption lacunae detection, osteoclast-like-cells were removed from sample surfaces by NaOCl solution. After dehydration and gold coating, samples were observed under SEM for resorption lacunae. To ensure that the TCP surface degradation was related to cellular activity rather than chemical degradation, control TCP disks were incubated in culture media with cells, but no RANKL was added to the media.

### 4.9. Ion Release

Ca^2+^ and (PO4)^3−^ release from TCP scaffolds was measured in ddH2O at 37 °C. Scaffolds were first weighed (250 mg) and then placed in 5 mL tubes (Eppendorf) containing 2.5 mL ddH_2_O. At different time points, 500 μL samples were collected for ion release measurements and replaced with fresh 500 μL aliquots of ddH_2_O. Calcium release was measured using a Quantichrom Calcium Assay Kit DICA-500 (BioAssay Systems, Rodgau, Germany) following the manufacturer’s protocol. In brief, a phenol-sulphonephthalein dye in the kit forms a very stable blue colored complex specifically with free calcium. The intensity of the color, measured at 612 nm, is directly proportional to the calcium concentration in the sample. Phosphate release was measured using a Quantichrom Phosphate Assay Kit DIPI-500 (BioAssay Systems, Rodgau, Germany) according to the manufacturer’s procedures. This method utilizes the dye malachite green and molybdate, which forms a stable colored complex specifically with inorganic phosphate. The intensity of the color, measured at 620 nm, is directly proportional to the phosphate concentration in the sample. All values were corrected for sample removal.

### 4.10. Specific Surface Evaluation

Specific surface evaluations were performed according to BET methodology [21] via absorption of N_2_ in an SA 3100 Surface Area and Pore Volume analyzer (Beckman Coulter, Orange County, CA, USA). Prior to analysis at −196 °C from 0.05 to 0.2 bar, samples were dried for 2 h at 180 °C.

### 4.11. Statistics

The primary analysis unit for in vivo experiments was one animal. For all parameters tested, treatment modalities were compared with a Kruskal–Wallis test, followed by pairwise comparison of treatment modalities with the Mann–Whitney test for dependent data (IBM SPSS v.19). P-values are displayed in the graphs and significance was set at a limit of *p* < 0.05. Data from eight different rabbits are presented for each group for the non-critical size defect. Values are reported in the text by mean ± standard deviation or displayed in graphs as median ± lower/upper quartile.

## 5. Conclusions

Having previously studied the influence of microarchitecture on osteoconduction, we here determined the effect of nanoarchitecture on osteoconduction and creeping substitution. We found microporosity indirectly enhances osteoconduction in wide-open porous TCP scaffolds by osteoinduction mediated by increased release of Ca^2+^ due to increased specific surface area. This is also the major aspect influencing non-cellular scaffold degradation. For osteoclastic scaffold resorption, however, grain size and grain boundary size could be the critical nanoarchitectural features. These findings are likely to pave the way for future developments in 3D-printed personalized TCP-based scaffolds, since optimized nanoarchitectural designs realized by defined post-processing conditions enable the tuning of bone ingrowth/regeneration capability and degradation velocity via defined dissolution and osteoclastic resorption characteristics.

## Figures and Tables

**Figure 1 ijms-21-09270-f001:**
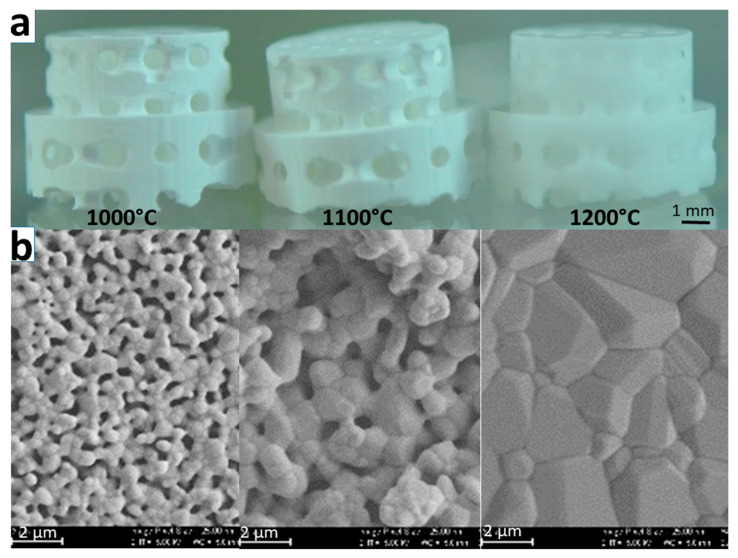
Scaffolds and SEM (scanning electron microscope) micrographs. Scaffolds are shown to maintain their microarchitecture after sintering at different temperatures (**a**). Scale in the upper panel is 1 mm. Sintering temperatures provided apply to both upper and lower panels. SEM micrographs of the respective scaffolds are displayed (**b**). Scale bars in the lower panels are 2 µm. Increase in grain size and decrease in microporosity (features of nanoarchitecture) with increasing temperature are illustrated.

**Figure 2 ijms-21-09270-f002:**
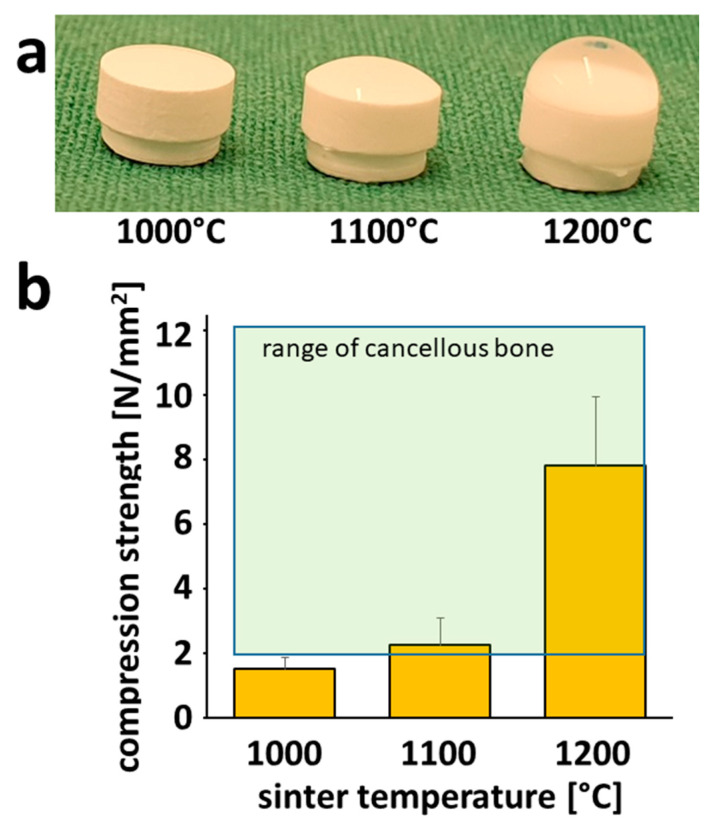
Microporosity and compression strength. Microporosity of test samples was evaluated by pipetting 30 µl of H_2_O onto a scaffold (**a**). Compression strength of partially sintered scaffolds (**b**). The range for cancellous bone (2–12 N/mm^2^) was taken from [22].

**Figure 3 ijms-21-09270-f003:**
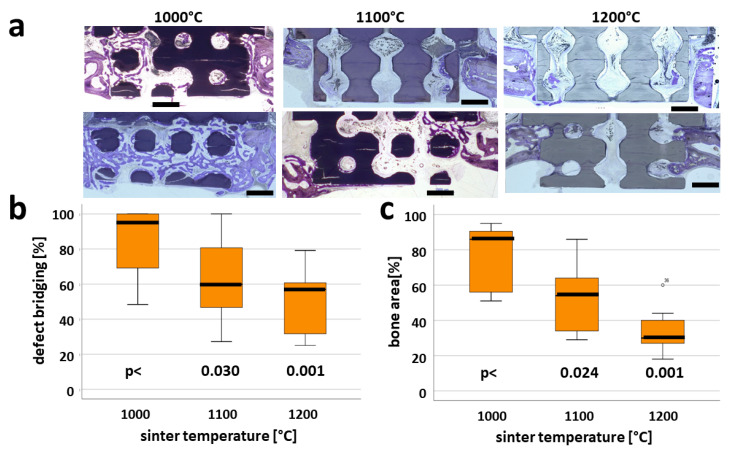
Microporosity-dependent osteoconduction and bone regeneration. Histological sections from the middle of the non-critical size defects treated with scaffolds sintered at 1000 °C, 1100 °C, or 1200 °C (**a**). For all scaffolds, two histological sections of the area of interest (AOI), which is the fraction of the scaffold inserted in the cranial defect, harvested 4 weeks postoperatively, are shown. Scale bars in black indicate 1 mm. Original magnifications were 100-fold. Bone appears as grayish purple to purple, and TCP as grayish to black. Histomorphometric parameters for bone regeneration in non-critical size defects 4 weeks postoperatively (**b**,**c**). Defect bridging (**b**), and formation of new bone (**c**) are significantly elevated in defects treated with scaffolds sintered at 1000 °C. Values are displayed as box plots ranging from the 25th (lower quartile) to the 75th (upper quartile) percentile, with the median displayed as a solid black line and whiskers extending to minimum and maximum values. Values outside the range of the box blot are shown as individual points. P-values are provided in the graphs.

**Figure 4 ijms-21-09270-f004:**
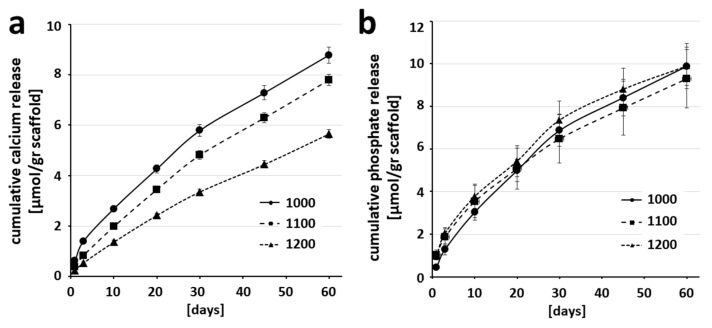
Sintering temperature-dependent Ca^2+^ and (PO4)^3−^ ion release.

**Figure 5 ijms-21-09270-f005:**
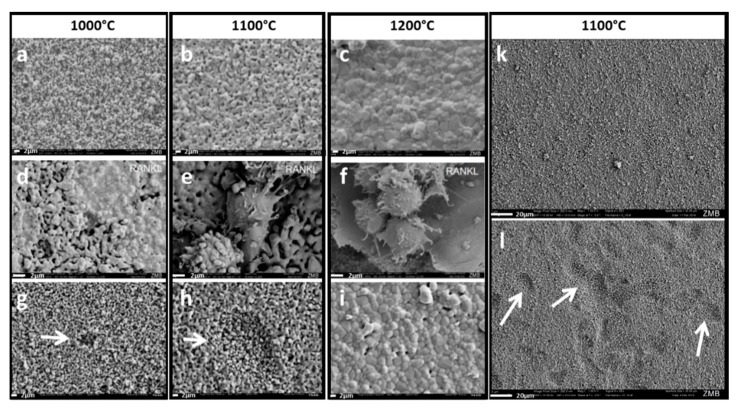
Sintering temperature-dependent osteoclastic scaffold degradation. Scanning electron microscopy was performed on scaffolds which underwent the entire procedure without cells (**a**–**c**)**,** scaffolds seeded with osteoclastic RAW264 cells and stimulated with RANKL (**d**–**f**), and on the same respective surfaces after removal of RANKL-stimulated osteoclastic RAW264 cells (**g,h,i**). Representative images of scaffold surfaces sintered at 1100 °C without exposure to RAW264 cells (**k**) and after exposure and lysis of osteoclastic RAW264-cells (**i**) is shown. The maximal sintering temperature is provided above each column. Scales in white are provided as 2 µm in a-i and 20 µm in (**k**,**l**).

**Figure 6 ijms-21-09270-f006:**
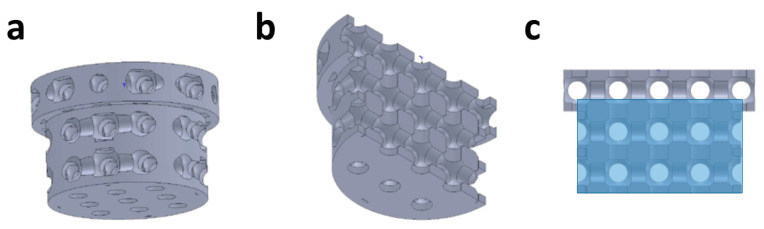
Micro and macroarchitecture of the scaffolds: (**a**) stepped cylindrical macroshape of the scaffolds, (**b**) middle section viewed from below, (**c**) front view of the middle section with area of interest (AOI) shaded in blue.

**Table 1 ijms-21-09270-t001:** Sintering temperature-dependent characteristics of scaffold nanoarchitecture.

Peak Sinter Temperature [°C]	Grain Size [µm]	Pore Diameter [µm]	Microporosity by Shrinkage [%]	Microporosity by Water Infiltration [%]	Surface [m^2^/g]
1000	0.71 ± 0.15	0.38 ± 0.16	38.84	39.84 ± 0.95	1.027
1100	1.24 ± 0.23	0.70 ± 0.24	22.82	21.41 ± 0.45	0.182
1200	3.08 ± 1.56	0.00 ± 0.00	0.00	0.98 ± 0.75	0.028

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
