# Peer review of "Microporosities in 3D-Printed Tricalcium-Phosphate-Based Bone Substitutes Enhance Osteoconduction and Affect Osteoclastic Resorption"

_ijms, 2020, doi:10.3390/ijms21239270_

Round 1

Reviewer 1 Report

Int. J Mol Scie: ijms- 1023713 Title: Microporosities in 3D-printed tri-calcium-phosphate-based bone substitutes enhance osteoconduction and affect osteoclastic resorption. 

The present study investigated the contribution of microporosity to the osteoconductivity of wide-open porous scaffolds and detected that the microporosity of TCP-based wide-open porous scaffolds is a key factor in osteoconduction. Additionally they could show that a lack of microporosity inhibited degradation of TCP-based scaffolds by osteoclasts. 

I read the study with great interest. The experiment design is clearly described, the data are presented appropriately and support the conclusions.

Comments

To asses osteoclastic resorption of the material, the resorption lacunae were detected by SEM documentation showing representative images of scaffold surfaces in Fig. 5. The authors could have also chosen an assay for detection and quantification of osteoclast activity like TRAcP staining (tartrate resistant acid phosphatase) assessing the resorptive activity.

For me, Figure 3a is not plausible and does not reflect the values of the histomorphometric analysis in 3b. In Fig. 3a, 1000°C, the upper half consists of grayish black TCP with 4-5 holes in the center while the lower half seems to consist of 4-8 round areas of TCP within a field of new bone. Fig. 3a 1100°C seems to show a perforated plate of TCP with only few bony areas in the lower half on the outside. Fig. 3 a 1200°C presents much more bony ingrowth than Fig. 3a 1100°C. Since 8 animals received 4 holes with each TCP group, you might want to show more than only one picture per group. 

What happened to the 4th hole? If this was left empty, it would be nice to compare with the 3 TCP scaffolds?

Author Response

Dear Reviewer 1.
First, I would like to thank you for critically reading our manuscript and for the suggestions. I hope, that the revised version meets now the requirements for an IJMS-publication.
Comments

To asses osteoclastic resorption of the material, the resorption lacunae were detected by SEM documentation showing representative images of scaffold surfaces in Fig. 5. The authors could have also chosen an assay for detection and quantification of osteoclast activity like TRAcP staining (tartrate resistant acid phosphatase) assessing the resorptive activity.
Thanks for this suggestion. Trap-activity is mainly used to study the differentiation of monocytes to osteoclasts. Here, however, we want to determine the degradation of material by osteoclasts. Therefore, we searched for resorption pits to show degradability by osteoclasts.

For me, Figure 3a is not plausible and does not reflect the values of the histomorphometric analysis in 3b. In Fig. 3a, 1000°C, the upper half consists of grayish black TCP with 4-5 holes in the center while the lower half seems to consist of 4-8 round areas of TCP within a field of new bone. Fig. 3a 1100°C seems to show a perforated plate of TCP with only few bony areas in the lower half on the outside. Fig. 3 a 1200°C presents much more bony ingrowth than Fig. 3a 1100°C. Since 8 animals received 4 holes with each TCP group, you might want to show more than only one picture per group.

What happened to the 4th hole? If this was left empty, it would be nice to compare with the 3 TCP scaffolds?
Thanks for this observations. In order to make it clearer we provided in the revised figure 3 for each sinter temperature, as suggested, two area of interests, which are the lower parts of the scaffolds inserted into the cranial defect. Differences in the distribution of holes or material are due to the sectioning to get the middle section. This cut through the middle of the scaffold is not predefined but at random, creating slightly different patterns.

Reviewer 2 Report

The manuscript by Weber and colleagues describes the impact of microporosities on ceramic, 3D printed scaffolds. The latter are evaluated in terms of porosity, mechanical features, calcium and phosphate release, osteoconductivity and osteoclastic resorption. The results are presented in a clear and rigorous manner, tracing new and interesting insights into the adjustment of scaffold microarchitecture to achieve the best in vivo performances.
I suggest only few revisions to strengthen the introduction section, as well as the materials and methods section.
Further comments are reported below.

Only in the discussion section it is mentioned the application field of this study (i.e. cranial defects). Even if the findings of this work are not only valid in this particular field, it is suggested to briefly discuss this point also in the introduction section.

In section 4.1, CAD model pictures of the scaffolds' architecture should be reported, even if they are already present in a database and/or in previous publications.

In section 4.5 representative images of AOIs for each sample type should be reported.

Lines 427-428: please add the number of cells seeded on each 3D printed scaffold.

Please check affiliation numbers, in particular affiliation 4 (no address reported)

Author Response

Dear Reviewer 2
First, I would like to thank you for critically reading our manuscript and for the suggestions. I hope that the revised version meets now the requirements for an IJMS-publication.

Only in the discussion section it is mentioned the application field of this study (i.e. cranial defects). Even if the findings of this work are not only valid in this particular field, it is suggested to briefly discuss this point also in the introduction section.
A section in the introduction was added about the cranial defect model.

In section 4.1, CAD model pictures of the scaffolds' architecture should be reported, even if they are already present in a database and/or in previous publications.
A figure 6a/b was added to illustrate the scaffold design.

In section 4.5 representative images of AOIs for each sample type should be reported.
A figure 6c was added to illustrate the AOI.

Lines 427-428: please add the number of cells seeded on each 3D printed scaffold.
The RAW264.7 cells were seeded at a density of 2.0 × 104 cells per scaffold. The scaffolds were placed at 37°C for 2h to allow cells to adhere followed by the addition of a complete growth medium. The growth medium was replaced with a fresh medium every 2 days to eliminate non-adherent cells.

Please check affiliation numbers, in particular affiliation 4 (no address reported)
Was deleted